# A Review of Cultural Practices for Botrytis Bunch Rot Management in New Zealand Vineyards

**DOI:** 10.3390/plants11213004

**Published:** 2022-11-07

**Authors:** Dion Charles Mundy, Philip Elmer, Peter Wood, Rob Agnew

**Affiliations:** 1The New Zealand Institute for Plant and Food Research Limited, P.O. Box 845, Blenheim 7240, New Zealand; 2Nelson Marlborough Institute of Technology Limited, Private Bag 19, Nelson 7042, New Zealand; 3The New Zealand Institute for Plant and Food Research Limited, Private Bag Waikato Mail Centre, Hamilton 3240, New Zealand; 4The New Zealand Institute for Plant and Food Research Limited, Private Bag 1401, Havelock North 4157, New Zealand

**Keywords:** botrytis bunch rot, mechanical thinning, cultural control, nutrient management, disease, grapes

## Abstract

Botrytis bunch rot of grapes (BBR) causes substantial crop and wine quality issues globally. Past and present foundations for BBR control are based upon synthetic fungicides and varying forms of canopy management. Many authors regard the continued dependence on fungicides as unsustainable and have urged greater deployment of cultural, biological and nutritional strategies. However, in contrast to organic wine production, the uptake of alternative strategies in conventional vineyards has been slow based on cost and perceived reliability issues. This review summarises research from many different wine growing regions in New Zealand with the aim of demonstrating how traditional and newly developed cultural control practices have cost-effectively reduced BBR. In addition to reviewing traditional cultural practices (e.g., leaf removal), mechanical tools are described that remove floral trash and mechanically shake the vines. Multi-omics has improved our knowledge of the underlying changes to grape berries after mechanical shaking. Exogenous applications of calcium may correct calcium deficiencies in the berry skin and reduce BBR but the outcome varies between cultivar and regions. Nitrogen aids in grapevine defence against BBR but remains a complex and difficult nutrient to manage. The sustainable growth of organics and The European Green Deal will stimulate researchers to evaluate new combinations of non-chemical BBR strategies in the next decade.

## 1. Introduction

*Botrytis cinerea* pers Fr (*B. cinerea*) is a globally important fungal plant pathogen responsible for many pre- and postharvest diseases of fruits, vegetables and field crops. Its global significance has been highlighted by many authors including Elad, Vivier, and Fillinger [1,2] and this pathogen is regarded as the second most important phytopathogenic fungus affecting crops world-wide [3]. In viticulture and table grape production the disease is commonly referred to as botrytis, grey mould or botrytis bunch rot (BBR) and this disease is arguably the most important that vineyard managers must manage across virtually all grapevine growing regions of the world. Precise global crop loss statistics are difficult to find but have been estimated at $US2B annually [4]. In cool-climate regions, such as New Zealand (NZ), direct crop losses of up to $NZ5000/ha have occurred especially in growing seasons favourable for BBR with additional costs of $NZ1500/ha for BBR control measures [5]. 

Conventional BBR control over the last 6-decades has relied heavily upon synthetic fungicides. However, soon after the introduction of the methyl benzimidazole carbamate (MBC) fungicides (FRAC Group 1) into a botrytis crop protection programme for glasshouse cyclamens, there was rapid development of resistance and a disease control failure [6]. In New Zealand, the widespread occurrence of benzimidazole resistance in populations of *B. cinerea* led to a major loss of efficacy in vineyards [7]. Since the 1960s there has been a re-occurring pattern beginning with the introduction of new synthetic chemistry (e.g., the dicarboximides, FRAC Group 2), followed by rapid uptake by vineyard managers, overuse in the vineyards over time, then reports of loss of disease control. Many authors have questioned the sustainability of this approach and highlighted the need for alternatives to synthetic fungicides. A summary of factors used to support the development of alternatives to fungicides is provided in Table 1. 

Biocontrol products, including antagonistic micro-organisms, natural products of mineral, plant or microbial origin and plant defence inducers have been recommended to reduce viticulture’s dependence upon synthetic fungicides for the last three decades [17,28,29]. However, many biocontrol authors would agree that uptake of biologically based strategies has been slow and reviews have identified variability in disease control and cost as significant barriers to uptake by vineyard managers [30]. While variability in disease control can be an issue, it has also been demonstrated that when the right combination of biocontrol products with different modes of action is well timed, BBR control was equal to that achieved with synthetic fungicide-based programmes [28,31,32]. Furthermore, some authors have reported biocontrol efficacy against BBR that exceeded 75% when compared with an untreated control [33]. In contrast, the use of some registered biocontrol products resulted in only 20% efficacy against BBR. The authors conceded that there were only 1–3 applications in a growing season, suggesting that there was insufficient protection of susceptible tissues at key growth stages in the growing season [18]. Across the New Zealand winegrowing regions, there is a gradient of temperature and rainfall from the warmer and wetter northern regions (Gisborne annual temperature 14.6 °C and rainfall 999 mm) to cooler and drier southern regions (Central Otago annual temperature 11.1 °C and rainfall 418 mm). Under these temperate growing conditions some seasons, especially in the warmer and wetter northern regions, can experience multiple *B. cinerea* infection periods and up to 10 biocontrol applications may be required in a growing season to achieve effective and reliable BBR control [28]. 

There is no doubt that variability of BBR control has been reported when synthetic fungicides are substituted with biocontrol products on a one-for-one basis. We suggest that this substitution paradox is not an adequate strategy and that the foundation for effective and sustainable biologically based BBR control strategies for now and into the future, must include the integration of biocontrol tools with scientifically validated cultural controls such as canopy management, debris removal and vine nutrition. Other reviews have covered chemical use and biological control agents; however, systems of vineyard management also contain a number of other measures used to control disease that do not involve spraying vines. We define cultural controls as the other methods of reducing the risk of BBR such as inoculum removal, changing the microclimate and indirectly increasing the vines defence system while not directly attacking the pathogen. In this review, we also summarise the epidemiology and ecology of *B. cinerea* in vineyards because this knowledge is essential to understanding the optimum deployment of cultural control operations. We then describe specific cultural control practices that have been used to manage BBR in vineyards.

## 2. Epidemiology of *B. cinerea*

Several publications have described the life cycle of *B. cinerea* in vineyards including Elmer and Michailides [4], Mundy, Agnew, and Wood [5], Gonzalez-Dominguez et al. [34], Fedele et al. [35]. The relative importance of each *B. cinerea* infection pathway has been the subject of scientific debate [36] because of the inherent variability between vineyards, varieties and regions. In summary, *B. cinerea* can survive and thrive in the vineyard as both a necrotrophic pathogen or as a free-living saprophyte. During the growing season, different vine tissues become susceptible to infection by *B. cinerea* with the potential for each tissue-type to contribute to BBR at harvest. Weather conditions from berry softening (the rise of sugars and change in colour (véraison)) to harvest often determine if infection potential translates to disease severity affecting yield at harvest. Post-harvest inoculum persists in tissue residues in the vineyard (e.g., grape bunch rachides in the canopy and rachides, leaf petioles, tendrils and cane lengths on the ground under the vine) [37], thereby increasing the potential *B. cinerea* inoculum risk in the spring of the next growing season.

Effective management of BBR takes place when the potential seasonal epidemic is interrupted to stop disease expression prior to harvest. Control measures that are applied either interrupt the *B. cinerea* lifecycle or change the susceptibility of the host tissue in the vine. Active management in New Zealand is normally achieved by applying multiple interventions at different growth stages targeting the disease or the vine. A range of BBR control measures that are acceptable for use in New Zealand are presented in Figure 1. 

### Botrytis Bunch Rot Management in New Zealand Vineyards

#### Background

In New Zealand’s wetter regions, BBR costs the wine industry up to $NZ5000/ha in direct crop loss and an additional $NZ1500/ha in disease management costs [41]. While it is possible to increase the number of interventions, this only results in higher production costs, which may not reduce BBR crop loss at every harvest date. Therefore, in most growing seasons, BBR management is a balancing act of cost versus predicted seasonal risk. Under New Zealand conditions, BBR at harvest may be the result of direct or indirect infection of the grape berry during the growing season. The berry can be directly infected at flowering (resulting in latent infections) and again after berry softening during ripening [4]. Within-vineyard epidemics can occur because of infection of other grape tissues between flowering and fruit ripening resulting in high spore numbers at susceptible stages of berry development under infection conditions for the pathogen [42]. Active management to prevent economic damage to the crop includes preventing physical damage to the berry skin that can increase berry susceptibility, and reducing bunch compactness to allow spray penetration, which reduces contact between bunch debris and berries. Removal of dead tissues from within the vineyard on which *B. cinerea* can grow is also a common practice as part of integrated disease management to reduce the build-up of spore pressure [5].

Most New Zealand vineyards routinely use a range of BBR management practices including removal of infected plant material at pruning, one to two applications of synthetic fungicides over flowering and some degree of canopy management such as vine trimming and leaf removal. For regions and sub-regions with greater potential risk of BBR or for premium high value varieties additional control measures such as mechanical thinning or green pruning may also be added [43]. Green pruning is the removal of shoots from the head of vines during the growing season to open the canopy; however, this practice is labour intensive to select which shoots to remove without reducing yield. Figure 1 provides a list of management practices that can be used for BBR management, which were current at the time of publication. The cultural controls options in Figure 1 will be discussed in more detail in later sections.

## 3. Cultural Control of BBR

### 3.1. Winter Vineyard Management

#### 3.1.1. Pruning

During winter pruning, the selection of new canes or spurs for the next season allows for the removal of diseased plant parts. For BBR management, the cutting out of bleached canes infected with *B. cinerea* or other pathogens is a key step in reducing inoculum for the coming season. In New Zealand, past research has shown the importance of removing other plant material as well as the canes. The importance of removing rachides [42] from the trellis and to a lesser extent tendrils [5,44] has been investigated. The aim of removing these tissues is to reduce spring inoculum in the canopy.

#### 3.1.2. Understory (Vineyard Floor) 

The understory or vineyard floor management includes the under-vine and inter-row area. In the past, at the end of the winter pruning season, some New Zealand vineyards removed and burned pruning material and other vine debris. With a change to managed cover crops in the inter-row instead of bare earth, prunings and other residual plant material are mechanically mulched for decomposition in situ. The inter-row environment provides a moist area with vegetative cover, which assists with decomposition of plant residues during the growing season; whereas the under-canopy area is often kept free of vegetation with the use of herbicides. Differences in the amount of trash between the under-canopy and inter-row have been observed, with significantly fewer rachides, petioles and cane-length trash detectable in the inter-row than the under-canopy in mid-summer [42]. The importance of different plant residues such as tendrils and rachides within the vineyard, as inoculum sources was reviewed in Mundy, Agnew, and Wood [5].

As crop residues are important potential inoculum sources, methods to reduce spore production have been studied. Internationally, weed control with mulches has been reported to reduce survival of *B. cinerea* on prunings [45]. The use of mulches and composts in boysenberries [46] and grapes [47,48,49] has been investigated in New Zealand. This method also provides other potential benefits to the crop, such as nutrient cycling, and is an acceptable practice for organic growers. We propose that competition with *B. cinerea* for space/nutrients by organisms specialised in decomposing dead tissue may be the mechanism of control when these systems are applied.

### 3.2. Canopy Management

#### 3.2.1. Bunch Trash (Debris) Removal

Over the period from flowering to pre-bunch closure (Modified Eichhorn-Lorenz growth stages 19–31), there is an abundance of senescent and necrotic floral tissues consisting of dead stamens, aborted flowers, aborted berries and calyptras [4,35,42,50]. Retention of these tissues in rapidly developing bunches can become problematic and *B. cinerea* is able to survive and effectively over-summer in a saprophytic state. Some clones such as UCD15 Chardonnay are especially susceptible to trash retention. Not surprisingly, the relationship between *B. cinerea* infection of bunch trash and subsequent BBR is well established in the literature, [50,51,52]. The impact of debris retention on BBR of the *Vitis* interspecific hybrid Vignoles was investigated between 2001 and 2005 by Hed, Ngugi, and Travis [53]. The accumulation of floral debris was found to contribute to BBR severity. In addition, the effect was greater in compact clusters compared with clusters that were classed as loose. In New Zealand, nine site years of data established significant correlations between *B. cinerea* infected bunch trash and BBR severity at harvest [54]. Therefore, a cultural control practice that can efficiently remove this significant source of *B. cinerea* inoculum has the potential to reduce BBR at harvest [36]. Between 2005 and 2007, the authors removed 100% of the bunch trash from developing bunches prior to bunch closure of Chardonnay in the Hawke’s Bay region of New Zealand with a compressed air gun. As a result of this treatment, BBR was reduced by 70% at harvest compared with the untreated control of no bunch trash removal. In separate studies, manual removal of flower debris from grape clusters (growth stage BBCH 73) with a small brush delayed the onset of BBR with the added benefit of increasing the ripening period in Pinot gris and Riesling in Luxembourg [52]. 

The air brush and paint brush techniques for bunch trash removal were valuable research tools but not practical at the vineyard scale. Fortunately, a tractor-mounted system (Collard pulsed air system, France) was introduced into New Zealand vineyards in 2007 and the authors observed up to 80% of aborted fruitlets and 60% of the flower calyptras were removed from developing grape bunches of Chardonnay in the Hawke’s Bay region of New Zealand. Importantly, the mechanical removal of bunch trash reduced *B. cinerea* inoculum potential in developing bunches by ~80%. In this industry commissioned research project the authors also found that there was evidence for regional and varietal differences in the efficacy of the Collard system, and variables such as moisture content of the bunch trash, vine vigour and bunch compactness were observed to affect the efficacy of bunch trash removal. Despite these drawbacks, the Collard pulsed air system integrated with leaf removal operations has now been widely adopted by New Zealand vineyard managers (~65%) as a BBR cultural control practice. 

#### 3.2.2. Leaf Removal

Globally, leaf removal in the fruit zone is one of the most effective tools for reducing the incidence and severity of BBR [55,56,57,58]. Leaf removal affects the fruit zone microclimate creating less favourable conditions for *B. cinerea* establishment. This is achieved by exposure of bunches to sunlight, increased air movement around bunches [59] and through significantly improved spray deposition of botryticides and biofungicides on bunches [11]. Canopy management trials incorporating leaf removal were conducted on Chenin blanc vines in California in 1984 and 1985 [55]. Disease conditions in 1985 were conducive for BBR development and leaf removal reduced the severity of BBR by 82%, compared with the untreated control. 

The first leaf removal studies in New Zealand were carried out in 1986–1987. Five trials in three regions utilising three grape varieties investigated the timing of leaf removal on fruit and wine composition, and viticultural aspects including BBR [56]. Reductions in BBR of between 41% and 86% were achieved depending on the timing of leaf removal. In addition, these trials evaluated the first field test of a tractor-mounted leaf removal machine in the Marlborough region of New Zealand.

Further and more extensive leaf removal studies were conducted in the Marlborough region on Sauvignon blanc [60] with a total of 15 site years of research (1996–1998). The results achieved an average reduction of BBR of 58% whilst achieving a 33% reduction in botryticides applied compared with a conventional calendar-based spray programme. Trials in Hawke’s Bay, New Zealand, over two seasons [54] and two varieties Chardonnay and Sauvignon blanc, demonstrated that the effect of leaf removal without botryticide sprays was similar to the standard botryticide programme and that the combination of both leaf removal and botryticide sprays reduced BBR by 97% and 94% in the Chardonnay and Sauvignon blanc, respectively. 

The timing of leaf removal is important as studies conducted by The New Zealand Institute for Plant and Food Research Limited (PFR) showed that late leaf removal (post véraison) did not reduce BBR. Würz et al. [61] demonstrated that leaf removal at full bloom and berries peppercorn-size was far more effective at reducing BBR than leaf removal at véraison or 15 days after véraison, compared with a control with no leaf removal.

Leaf removal between flowering and véraison with fruit exposure of 75–90% reduced herbaceous characters in Sauvignon blanc [56,62]. Most vineyards in the Marlborough region of New Zealand undertake mechanical leaf removal. However, in order to retain the unique herbaceous characters for which Marlborough Sauvignon blanc is well known, vineyards in this region generally do not allow fruit exposure of greater than ~40%. The density of these typical canopies has been well defined [63] and can now be modelled if canopy systems are to be changed or modified [64]. The implications of this lower scale of leaf removal means that grape bunches in these more dense canopies were more susceptible to BBR. Decreased BBR incidence and alteration of wine flavours following leaf removal treatments have also been reported for other varieties [65]. With the wine style of New Zealand Sauvignon blanc negatively affected by excess leaf removal, a new cultural control practice to mitigate BBR has become well adopted in the Marlborough wine growing region and this method of cultural control is referred to as mechanical shaking. How it evolved into a new tool for BBR control is described in more detail in the next section.

#### 3.2.3. Mechanical Thinning 

Mechanical thinning in Marlborough, New Zealand, was not originally intended for BBR control. Rather, disease scoring was included in the project as the team was concerned that the practice of applying mechanical thinning to vines may increase the risk of *B. cinerea* infection due to increased damage to developing berries. When replicated experiments of mechanical thinning were conducted, the fruit had consistently lower BBR compared with the control as well as effective fruit removal for yield control. 

In vineyards, fruit thinning with a mechanical grape harvester is often used to manage yields so that grapes mature to optimum soluble solids and ripeness, at any site and across seasons [66]. The control of vine yield is important for successful high-quality wine production and can have direct and indirect effects on the observed BBR at harvest. As a result of the temperate climate in New Zealand, at least two-fold differences in seasonal yields of Sauvignon blanc and Chardonnay have been recorded [67] so that thinning may not be required in all seasons. This variability, together with seasonal differences in flowering dates and temperature, affect the likelihood of fruit achieving adequate ripeness in any year. Thinning is a technique where growers remove fruit to allow the remaining berries to develop to the desired ripeness before the end of the growing season. Hence, strategic crop manipulation to achieve target yields will not be required every year but when used may remove a variable percentage of crop. Growers generally prefer to undertake thinning after fruit set, when potential yields have been determined for the coming harvest and yield components such as increased berry size cannot compensate for the yield reduction [66,68,69]. However, the later in the season that thinning takes place the more costly the yield removal. 

Mechanical thinning of grapes was first used on Concord juice grapes in the USA [70] and more recently in both Australia and Europe on red grapes [65,71,72,73]. Other reports on using mechanical methods of crop load management have focused on reducing labour cost [74,75] and have not reported interactions between thinning and observed disease. Following field observations of reduced BBR at harvest on vines thinned by shaking, Mundy et al. [76] observed that: (1) mechanical thinning changed the bunch structure by removing parts of the bunch, resulting in a more open bunch with lower disease risk; (2) trauma of the berries during mechanical thinning brought about changes such as induced resistance or increased skin thickness resulting in reduced susceptibility of berries to infection; and (3) mechanical thinning action removed debris or trash from the bunches, resulting in reduced sources of disease. 

While the method was first developed for thinning crops, and terms such as heavy and light mechanical thinning have been used by the industry to reflect how much removal of fruit was achieved, the shaking of vines specifically for botrytis control is now practiced by some growers. The term “mechanical shaking for botrytis control” is sometimes shortened to “mechanical shaking”. This technique uses less beater rods in the machine and lower settings to induce a potential reduction in BBR in the vine while removing less than 5% of yield. This contrasts to the experimental yield reductions reported for light mechanical thinning (22–23%) and heavy mechanical thinning (26–43%) [76].

The aim of mechanical shaking was not to replace early season management options such as chemical sprays but to complement them with a mid-season option to reduce BBR risk at sites with historically higher disease pressure or higher value yields. Currently in New Zealand, available chemical BBR controls are primarily focused on early season actions by necessity, because late-season chemical options are limited to avoid residues in wine. 

##### Bunch Structure

Field observations suggested that a more open bunch can reduce the risk of BBR at harvest. In table grapes plant hormones and other methods have been used to change bunch shape in order to create more open bunches and to reduce disease at harvest and post-harvest. Clonal differences in bunch compactness have been associated with differences in *B. cinerea* susceptibility [77]. The concept is that a more open bunch: does not hold debris; allows better air movement for drying; and improves spray penetration allowing better control of any *B. cinerea* on non-berry tissues between flowering and fruit ripening.

Bunch openness experiments have been conducted in Marlborough, New Zealand, to see if opening the bunch for wine grapes may also reduce the risk of BBR at harvest [78]. In order to measure bunch openness, Mundy et al. [76] calculated bunch volume from the longest and widest points using the established method of comparing bunches between cultivars and seasons [79]. This bunch openness method was also used in experiments to change bunch shape with plant hormones and inhibitors to reduce the risk of BBR [78]. The mechanical thinning study [76] recorded significant differences in bunch volume between bunches from the control vines compared with the heavy shaking. The Mundy et al. [78] hormone treatments also had significantly more open bunches for some treatments. However, the main finding between the two studies was the consistent reduction of BBR disease severity in the mechanical shaking (2010 and 2011 vintages) compared with no significant differences for the same vintages when only the bunch openness was increased. Both studies were conducted in Marlborough on Sauvignon blanc vines, with the hormone studies conducted at a different vineyard site but during the same weather events. The observations from the two studies raise the question, “Can commercial reduction of BBR at harvest be achieved by increasing bunch openness alone?” Additional experiments, possibly including metabolomics, would be required to test this idea as the current methods to change bunch openness in existing clones also change berry and vine metabolism.

##### Berry Susceptibility

If the bunch openness as a result of mechanical shaking does not fully account for the observed reduction in BBR severity at harvest then does the trauma of shaking vines at pre-bunch closure change the susceptibility of individual berries to infection at harvest? The Mundy et al. [76] publication used a berry susceptibility assay previously published [80] to investigate the question of trauma-induced defence. The berry bioassay indicated that berries from heavily mechanically thinned vines that were not wounded were less susceptible to *B. cinerea* infection following artificial inoculation than wounded berries. The assay was conducted under ideal conditions for the fungus with incubation for seven days at 20 °C and 100% humidly and with a high spore loading (20 µL 1–5 × 10^4^ spores per berry); however, a reduction in susceptibility of 20% was observed under these conditions. These conditions at harvest would not be expected in New Zealand for seven days as those temperatures and humidity are not expected in the field in autumn. Published literature has indicated that environmental changes (e.g., increased fruit exposure to light and air flow) can increase the thickness of the epidermis of grape berries, which in turn may be associated with a reduction in the susceptibility of berries to *B. cinerea* [81]. In experiments applying mechanical thinning [82], the susceptibility of wounded berries following heavy machine thinning was observed to have reduced susceptibility compared with the control. It can be inferred from the low incidence of disease on non-inoculated control berries (either wounded or not) in the same experiment that latent infection was not commonly present in the berries used in the assay. The reduced susceptibility of individual inoculated berries without wounding in the heavy machine treatment has been further investigated in subsequent experiments to determine what chemical changes to primary metabolism of green berry physiology result from machine thinning treatments [82]. The researchers observed distinct changes in primary berry metabolism of green berries at the time of shaking as well as significant changes in berry secondary metabolism as fruit softened later in development. Shaken vines accumulated phenolic compounds as well as amino and fatty acids near the skin of the berry. While possible changes in berry metabolism may result in less disease at harvest, the wine industry has been concerned that such changes in chemistry could influence the wine made from the grapes that have been mechanically shaken. In related studies, late-season trauma to berries during harvesting was investigated to determine if pathogenesis-related proteins and phenolic extraction of juice increased with mechanical processing and harvesting [83]. Both sets of compounds can be a problem during production of white wines; if present they require additional winemaking treatment to stop haze-forming in the finished wine. Although individual wine companies in New Zealand have conducted commercial scale wine making from vines with and without mechanical shaking, these data have not been published. Chemical and sensory evaluation of wines produced from berries subject to mechanical shaking trauma at pre-bunch closure is required. Currently, thresholds of target compounds to reduce susceptibility have not been determined and the trauma of mechanical shaking was not sufficient to reduce susceptibility of all berries exposed to the treatment. Significant reductions in field severity of disease have been observed when the stronger mechanical thinning treatments have been applied, resulting in reductions of disease by 50%, which could not be fully explained by a 20% reduction in susceptibility in the berry assay under ideal conditions. 

##### Debris

As noted above, it is possible to reduce the risk of BBR at harvest when all debris is removed from the bunch. However, the mechanical thinning did not remove all debris from the bunch even at the heavy shaking setting. More recent studies in the Marlborough region have investigated a two-factor experimental design with and without mechanical thinning and with and without a Collard pulsed air system. This five-year investigation is due for completion in April 2023 and preliminary results indicate an additive effect of both methods for BBR management (lighter shaking than for thinning) and the use of the Collard system compared with either treatment applied alone. Continued studies of this type will determine if the reduction in debris or *B. cinerea*-infected debris per bunch accounts for the decreased harvest BBR incidence and severity observed in vines that are mechanically thinned or shaken for BBR management.

International research has shown that flower debris and latent infections can be important for BBR epidemics [52,84]. The viticulture industry already uses other methods such as mechanical blowers to try and remove debris from grape bunches. In general, the importance of debris in the epidemic of BBR within crops has been discussed in detail [4]. Under New Zealand grape production conditions, studies have shown the importance of debris as a source of spores for later infection of the berry when it is susceptible [42,44]. While reports of BBR in mechanical thinning for crop control are not numerous, the Mundy et al. [76] results are consistent with international research looking at red grapes [73]. In that study, Tardaguila et al. [73] were interested in the effects of thinning on physiological measurements and yield. BBR disease incidence was recorded as they assessed bunches so that those bunches with BBR could be excluded from calculations of yield. As the Tardaguila et al. [73] study did not have a pathology component, disease severity was not recorded. In the Tardaguila et al. [73] study, all mechanical treatments reduced incidence of BBR compared with the control regardless of timing or method of yield reduction. The Mundy et al. [76] investigation was instigated on the assumption that mechanical thinning would result in more disease due to infection of debris resulting from the trauma. The results indicated less disease; however, debris incubations of material from vines thinned before *B. cinerea* infection periods were not included in the experiment’s reports as field infection directly following treatment did not occur during the three study seasons. Investigations of bunch debris at the time of thinning would determine if the fewer debris pieces in the thinned bunches were due to direct shaking of the bunch during the thinning or the removal of individual berries allowing debris to fall from the bunch over time before harvest. New methods of detecting spores in debris and potential spore production [85,86] from the dead plant material in the bunch could be used for a detailed study of the importance of this pathway of infection in the epidemic of vines receiving mechanical shaking or control treatments. If all debris cannot be removed by mechanical shaking and or the use of devices such as the Collard pulsed air system, then other methods to suppress inoculum within the trash such as targeted fungicides or biological control agents [35] may need to be considered if this pathway is important for harvest disease.

While practices such as leaf removal and mechanical shaking can modify grape berry morphology and biochemistry to reduce *B. cinerea* infection and development, other vineyard factors such as vine nutrition can also be important.

## 4. Management of Nutrients

In this section we review how the management of two selected nutrients, calcium and nitrogen, might be manipulated to reduce BBR risk in the vineyard.

### 4.1. Calcium

The importance of the calcium ion (Ca^2+^) as a structural component in plant cell walls and as a signaling agent involved in the regulation of diverse cellular functions has been described by many authors (e.g., Hocking et al. [87], André et al. [88]). It is also well reported that fruits deficient in calcium are also more susceptible to physiological and pathological decay [89]. In order to understand more about calcium and its importance in grape berries, it is important to understand the underlying physiology of grape berries and the changes that occur at véraison. 

#### 4.1.1. Calcium-Grape Berry Water Flow Relationships 

The transpiration rate of grape berries was reported to be drastically reduced post-véraison in wine grapes [90]. In addition, the breakdown of mesophyll cell membranes (cell death) commenced at the end of véraison in Shiraz, Chardonnay, Cabernet sauvignon and Nebbiolo. The impact of this change was a reduction of the water potential gradient between the berry and the stem xylem, thus affecting water flow rate to the berry [91,92]. Calcium-carrying water flow rates into berries of Chardonnay and Thompson Seedless were relatively high pre-véraison, but declined post-véraison, potentially reducing calcium transport into these berries. In comparison, Shiraz water flow rates post-véraison were maintained at pre-véraison rates [93], suggesting that different varieties of wine grapes may differ in their capacity to deliver calcium to the ripening berry. These physiological processes suggest that calcium deficiency in some varieties of wine grapes is likely to be more common than previously thought. Overall, the accumulated evidence suggests that it is the post-véraison stage of berry development that offers the greatest opportunity to artificially modify berry calcium content. 

Prior to 2010, calcium research on BBR was directed at the relationship between the calcium content of grape berry skins and the direct effects of exogenous calcium on conidial germination and growth of *B. cinerea*. In the next two sections we briefly review how calcium interacts with the grape berry and *B. cinerea* at the physiological, biochemical and metabonomic scales. 

#### 4.1.2. Direct Effects on *B. cinerea*

Calcium chloride reduced the extra-cellular polygalacturonase activity of *B. cinerea* by up to 90% [94,95] and in separate studies calcium was also reported to directly affect *B*. *cinerea* conidia and hyphae causing conidial malformation and cytoplasmic disorganisation [2]. In another study, calcium chloride inhibited *B. cinerea* spore germination and mycelial growth under laboratory conditions [96].

#### 4.1.3. Indirect Effects on *B. cinerea*

Significant negative correlations between the calcium content of grape skin cell walls and susceptibility to enzymatic digestion by *B. cinerea* were reported by Chardonnet and Doneche [97]. Exogenous calcium application to grape berries was found to trigger the host antioxidant defence response against *B. cinerea* [2]. In separate studies the effect of exogenous calcium applications on grape berry, physiological, metabolic, transcriptional and microbial status of Vinhao (a Portuguese wine grape variety), were investigated. The authors reported that calcium reduced fruit cracking, restructured the grape berry waxes and changed the relative abundance of filamentous fungi on the berry surface and reduced postharvest storage decay [98]. When vines of Vinhao were sprayed with a 2% solution (*w*/*v*) of calcium chloride and 0.1% (*v*/*v*) Silwett L-77 as a surfactant several biochemical and transcriptional modifications were measured. The authors found that in the calcium-treated vines there was overexpression of the cell wall and pathogen defence-related genes pectin methlyesterase (PME), polygalacturonase-inhibiting protein (PGIP), serine protease inhibitor (PIN) and pathogenesis-related protein 1 (PR1). It was hypothesised that these outcomes most likely contributed to the observed reduction in fruit rot [89]. 

#### 4.1.4. Practical Use of Calcium in the Vineyard

Historically, the effect of exogenous calcium applications on BBR in grape berries has been variable. This variability may be due to the timing of exogenous calcium sprays, which can be critical. For example, two pre-véraison sprays to the table grape variety Italia were more effective than two post-véraison sprays at reducing BBR during storage [99]. Two applications of 1% calcium chloride (*w*/*v*) 90 and 30 days before harvest of Italia table grapes were most effective in reducing *B. cinerea* field rots, while two applications 21 and 5 days before harvest were most effective in reducing storage rots compared with untreated controls [95]. 

In our unpublished studies on wine grapes, the foliar application of 40 mM calcium chloride to Chardonnay, Pinot gris, Pinot noir and Sauvignon blanc wine grapes demonstrated that at least four applications were necessary to maximise calcium concentration in the berry skin. Furthermore, the calcium chloride that was applied from véraison resulted in higher concentrations of calcium in the skin than applications made earlier between berries at pea sized and véraison. Calcium uptake into the skin was not affected by relative humidity under New Zealand growing conditions. In a separate trial, the application of calcium chloride reduced BBR development in four out of five field experiments in commercial Chardonnay vineyards in the Hawke’s Bay wine growing region of New Zealand. 

### 4.2. Nitrogen and Berry Diseases

A comprehensive review on vineyard nitrogen management practices and grapevine physiology has been published [100] including a section on the negative effects on wine making as a result of low or high juice nitrogen. This review points to the complex interactions that occur in a vine when nitrogen is added and how these changes can be different in vines with low, moderate or high, nitrogen reserves. Some of the factors that can change in a vine due to nitrogen that are important for BBR are innate berry defence mechanisms, such as skin thickness or metabolic pathways. Changes in vine nitrogen can also indirectly affect the suitability of the micro-environment for infection by pathogens, such as bunch architecture, or the vegetative growth form of the vine. Some viticultural practices, such as leaf plucking, the trellis system or irrigation, can alter the micro-environment and therefore indirectly influence potential infection risk. 

Nitrogen (in the form of amino acids) is a key building block for many of the grapevine defence compounds. When amino acid pools within the vine are limited, defence compound pathways can be inhibited. Berry defence mechanisms that successfully prevent infection by *B. cinerea* are governed by genes that provide physical and chemical barriers to infection. BBR manifests in grapes when *B. cinerea* has the genetic ability to overcome the vine’s defence systems under certain conditions. In New Zealand’s principal grape-growing areas, the main disease affecting berries pre-véraison is powdery mildew, and post-véraison it is BBR. Other bunch rots are sometimes also detected at harvest but are normally the result of prior infection with BBR or physical damage to the berry such as trimming or bird damage. The change in berry susceptibility to these two diseases is likely a result of chemical changes in the tissue. Robinson, Jacobs, and Dry [101] discussed the possibility that the constitutive expression of IV chitinase post véraison may be responsible for berry resistance to powdery mildew following véraison. The véraison changes in the berry also seem to have an effect on the susceptibility of fruit to infection by *B. cinerea*. 

#### 4.2.1. Inherent Berry Defence Mechanisms

Successful infection of a grape berry by *B. cinerea* requires the activation of a number of genes within the plant pathogen to overcome the plant’s defences. These genes are under molecular control and able to be expressed when the berry is susceptible. Snoeijers et al. [102] outlined the range of genes that allow *B. cinerea* to successfully infect the grape. The nit gene system allows the use of secondary nitrogen metabolites in the berry as a nitrogen source for the fungus. The mpg1 gene mediates appressorium formation leading to cell wall penetration. The vir and path genes are induced by low nutrient status of plant cells, resulting in upregulation of the fungal metabolic pathways and growth.

*B. cinerea* can also be a successful pathogen by avoiding or disabling the grapevine’s defence responses. Robert et al. [103] reported on the inducible chitinase gene expression in leaves and fruit of grapes. They suggested that chitinases are differentially expressed depending either on the developmental stage of berries or on the type of infecting pathogen. They reported that no Class I and III chitinase expression was detected at any development stage of berries in the absence of plant pathogens. They also reported that *B. cinerea* did not induce Class III chitinase in leaves, while other pathogens did. These findings suggest that *B. cinerea* is equipped with some sort of mechanism that precludes expression of Class III chitinase in leaves: this provides it with a competitive advantage over other pathogens infecting grape leaves. As we learn more about the metabolism of both *B. cinerea* and grapevines, other plant host interactions will be elucidated. Recently published New Zealand work provides details of how berry nitrogen metabolism can be changed by management practices and how these changes may reduce field-observed BBR [76,82].

#### 4.2.2. Direct and Indirect Influence of Nitrogen

The application of nitrogen to vines in the field can influence the susceptibility of berries to infection either directly or indirectly. Changes that result in differences in the skin or metabolic pathways of the berry will directly affect the susceptibility. These differences may be independent of indirect influences or may interact with them.

Physical and morphological factors of grape berry cuticle development and susceptibility to BBR are likely linked. Commenil, Brunet, and Audran [104] reported on the changes in the thickness of cuticle during berry maturation for three clones of Pinot noir, with different susceptibilities to BBR. A higher degree of cracking of the cuticle was detected on the clone that was most susceptible to BBR, but no other physiological or morphological modifications of the cuticle were detected during development. More recently, observations of berry skin using an impedance meter referenced to cuticle thickness have been able to contribute to phenotyping of botrytis-resilient wine grape varieties [105]. Keller, Arnink, and Hrazdina [106] reported that the weight of skins and the skin to berry ratio was decreased with high nitrogen applied as NH_4_NO_3_ at bloom. Hence, the addition of nitrogen to vines may have effects on skin and cuticle development, and should be investigated in more detail. The management of vineyard nitrogen remains a complex and difficult operation as too much nitrogen can be a problem for winemaking (with protein haze), as can too little (stuck ferments) [107]. Within a vineyard, management of vine nitrogen may have to include consideration of how additional nitrogen may reduce the berry skin thickness and increase susceptibility.

Even if *B. cinerea* penetrates the physical barrier of the berry skin, the plant still has a number of chemical defence pathways, many of which rely on amino acids as precursors. Increases in berry amino acid concentration will therefore influence the berry’s potential to produce these chemical defences.

Keller and Hrazdina [108] investigated the profile of anthocyanins and how they were altered by increased nitrogen at bloom and light levels at véraison. The formation of phenolics closely paralleled changes in sugar accumulation, increasing exponentially in weeks 1–3, then slowing to reach a maximum at 5 weeks’ post véraison. Low nitrogen treatments enhanced phenolic formation, particularly flavonol glycoside development at the beginning of ripening. Whilst sun exposure was similar for all treatments, the low nitrogen treatment bunches did have fewer berries. By harvest these differences were smaller for all classes of phenolics as long as véraison was not light limited. Light at véraison had a greater effect on anthocyanins development than bloom nitrogen. Vine nitrogen had a direct influence on which individual pigments formed in the skin, in addition to the indirect effect due to the stimulation of vegetative growth and fruit set. Low light at véraison and high nitrogen resulted in a significant colour shift more towards red. The same Cabernet sauvignon grapes grown in full sun and low nitrogen status would produce a more crimson to purple hue. 

Keller, Steel, and Creasy [109] also reported differences in berry nitrogen-rich chemistry as a result of the light environment but did not investigate the effects of adding nitrogen. They noted that in green berries stilbenes (phenolic phytoalexin compounds) stop or delay the progress of invasion by *B. cinerea*. The skins of berries grown in full light accumulated more of these phenols than shaded berries. Photosynthetically active leaves producing sucrose are required to produce energy and are precursors for stilbene production. The authors recommended undisturbed canopy development from budbreak until bloom to ensure that enough photosynthesising leaves are present to avoid stress that might negatively affect production of these plant defence compounds. This suggestion would prevent trimming or leaf plucking before flowering.

Bezier, Lambert, and Baillieul [110] found that phenylalanine-lyase (PAL), stilbene synthase, acidic chitinase and polygalacturonase inhibitor proteins (all plant defence compounds) were detected in infected leaves 6 h after inoculation with *B. cinerea*. The activation of these genes did not stop *B. cinerea* spread within the tissue. The acidic chitinase VCH3 was not detected in berries (not a class IV). PAL, stilbene synthase, basic chitinase, and polygalacturonase inhibitor proteins were detected in highest quantities late in the infection process. Detection was of gene induction, not of protein production. In other plant systems, expression of polygalacturonase inhibitor proteins has been shown to reduce the growth of *B. cinerea*. The defence mechanisms of grapevines in response to pathogens has been fully reviewed in Monteiro et al. [111]. 

Many New Zealand growers have noted that low nitrogen vineyard sites have low BBR at harvest, and have attributed this low disease severity to low berry yeast available nitrogen (YAN) values. As low berry YAN can be a problem during fermentation, testing this observation was important to determine if a causal relationship existed between berry YAN (i.e., ammonium and amino acid content) and berry susceptibility.

Experiments conducted to investigate if the increased presence of nitrogen in the berry was directly related to susceptibility of the berry to infection did not show a direct relationship, with changes in sugar content of the berry during ripening having a more significant effect on infection [80]. It is possible that the indirect effects such as tighter bunches due to improved fruit set, and dense canopy due to increased vegetative growth, are the reasons for increased observed BBR at high nitrogen vineyard sites. The possible direct and indirect effects of nitrogen on BBR are discussed fully by Mundy [112]. In summary, the 2008 review discusses direct and indirect effects of nitrogen on berry infection rates. Direct effects discussed are berry skin physical properties and plant defence molecules.

The work of some researchers to combine additional nitrogen with green or summer pruning to successfully offset increased disease risk is also discussed by Mundy [112].

Many tissues within the plant require nitrogen. Therefore, the addition of nitrogen to the soil does not always result in direct changes to the nitrogen content of the fruit. Researchers report a range of nitrogen plant interactions that can indirectly influence the susceptibility of berries to *B. cinerea*. Keller, Kunner, and Carmo Vasconcelos [113] found that nitrogen increased bunch stem necrosis, BBR and yield. The severity of *B. cinerea* infection was correlated with the number of berries in the bunch and with bunch weight. These results are consistent with the findings of Christensen et al. [114] who reported that more rots were detected in vines with higher nitrogen treatment and véraison application. They defined rot as berries with four or more adjoining berries showing decay. They also reported a significant increase in yields produced by Grenache vines in these treatments. These results are also consistent with the discussion above of more open bunches (less compact) being a possible mechanism for reduced BBR following mechanical shaking.

Spayd et al. [115] concluded that increased canopy size and delayed harvest with increasing nitrogen fertilisation probably increased the incidence of infection of *B. cinerea*. However, as incidence was not determined in that study, the conclusion was based on non-quantitative observations. The phenol concentrations of the Riesling grapes investigated decreased quadratically with increased nitrogen fertilisation. This decrease may have been due to heavy canopy development, with nitrogen fertilisation resulting in increased fruit shading. The maximum recommended nitrogen application to well-managed trickle-irrigated vines was 56 kg N/ha. Hilbert et al. [116] also reported an increased leaf area with increased nitrogen, which can indirectly increase susceptibility by changing the micro-environment.

The vegetative responses of vines to nitrogen are likely to indirectly influence the risk of *B. cinerea* infection. Therefore, the timing and rate of any nitrogen addition should be controlled to reduce possible negative effects. 

Fruit set can be increased at sites that are low in nitrogen by providing the nutrient before flowering. While increased fruit set does not directly increase BBR in the field, it does often delay ripening (increased yield requires more sugars to reach the same concentration per berry) as well as making the berry to berry movement of disease possible when skins touch, without the need for free water. Tight bunches can also be very hard to spray and can trap flower parts that can become infected with *B. cinerea* and increase infection at harvest. The New Zealand industry has investigated methods to keep a more open bunch structure, and hence reduce the risk of disease spread. These methods include, but are not limited to, flowering sprays [78] and mechanicalthinning [76]. 

While adding nitrogen to vines is known to have a direct effect on canopy growth by increasing vegetative production and hence an indirect effect on BBR by producing a dense canopy with a microclimate more conducive for disease, measures can be taken to offset these changes. The main method reported internationally for offsetting increased risk of disease due to changed microclimate around the bunch is the use of leaf removal and summer pruning. In some regions of New Zealand such as Gisborne and Hawke’s Bay this method is the mainstay of an integrated disease management system of sprays and cultural controls as discussed above. The integrated control approach uses the knowledge of the disease life cycle to target different points of possible control. Often the use of one or more tools will depend on the cost/benefit the grower perceives from adding another control method into the disease management system [5]. However, for Sauvignon blanc grapes grown in Marlborough, removing most of the leaves in and around the bunches is not an option, as these leaves are an important factor in the production of the unique style of wine for which the region is famous. As Sauvignon blanc wine is the flagship of the New Zealand industry and the style of wine that generates most export value, disease management of these grapes cannot include heavy leaf removal from the shoots bearing fruit. Light leaf removal in the fruit zone on one side of the canopy is an option.

#### 4.2.3. Summary—Interactions between Nitrogen, Grape Berry Quality and Pathogen Susceptibility 

Nitrogen is an important macronutrient for the growth and defence of grapevines. The nitrogen and carbon status of the vine will determine the fruit set, vegetative growth and the chemical compounds present in the fruit. Many of the nitrogen-related interactions between the host and pathogen are indirect effects resulting from changes in micro-environment and crop load. There are a number of *B. cinerea* genes that allow growth at low cell nutrient contents, so an increase in cell nitrogen could favour the host plant. Juice nitrogen also affects wine quality, with optimum berry nitrogen tending to produce wines with greater flavour and aroma [117,118]. Therefore, while nitrogen limitation may be one method of controlling vegetative growth and bunch architecture, and hence indirectly reducing susceptibility to BBR, this may also reduce the quality of the wine produced.

## 5. Conclusions 

Botrytis bunch rot of grapes continues to cause substantial crop and wine quality issues globally. The foundation for effective disease control has been based upon applications of synthetic fungicides at key phenological growth stages, combined with varying degrees of canopy management. Many authors have questioned the sustainability of viticulture’s reliance on fungicides and have urged the greater deployment of cultural, biological and nutrition based strategies in vineyards [28,33]. In organic wine production, where synthetic fungicides are not allowed, the integration of these multiple strategies ensures effective BBR control and the production of premium quality wines keenly sought after by consumers. In contrast, the uptake of these strategies in conventional wine production has been arguably slow because of concerns about cost and reliability [18,30]. 

This review summarised key scientific findings from many different wine growing regions with the aim of demonstrating how newly developed and traditional cultural control practices have the potential to successfully suppress BBR epidemics. Winter vineyard pruning aims to reduce *B. cinerea* inoculum potential for infection of floral debris in the spring, and the benefits of under-vine mulching were described [5]. The removal of bunch trash to reduce *B. cinerea* inoculum within the bunch has received a lot of attention and there are mechanical tools that cost-effectively remove this important source of inoculum. Leaf removal, a cultural practice that has been advocated since 1927 (Elad Pers. Comm.) effectively reduced BBR by at least 50% and has become an accepted practice in many New Zealand vineyards in the North Island wine growing regions where BBR risk on average is greater than the South Island regions. Potential adverse effects on specific wine quality parameters has slowed uptake of this technology in premium Sauvignon blanc vineyards and prompted testing of alternative mechanical systems. In the last decade, the innovative use of mechanical grape harvesters to shake the vines at selected growth stages has resulted in significant cost-effective reductions of BBR [76] and this practice represents a breakthrough in non-chemical management of BBR in the last decade. Biochemical and multi-omics investigations have described some of the underlying changes to grape berries that account for reduced susceptibility to *B. cinerea* with further studies underway.

Manipulating the nutritional content of the grape berry using exogenous calcium applications and nitrogen were described as additional tools that have the potential to reduce BBR epidemics. Three decades of basic and applied research supports the exogenous application of calcium to developing bunches, with the greatest benefits reported in cultivars with a calcium deficiency in the berry skin. The authors believe that this practice has the potential to complement, rather than replace existing cultural tools. Nitrogen is an important macronutrient for the growth and defence of grapevines. However, the management of vineyard nitrogen remains a complex and difficult process as too much nitrogen can be problematic for winemaking, as can too little [107]. Hence, targeted management of nitrogen for vines may have to include consideration of how additional nitrogen may influence the berry skin in a specific cultivar and vineyard location.

Large gaps in research exist where new and traditional cultural control practices are integrated with biologically based tools for reliable and cost-effective suppression of BBR in a wide range of vineyards and wine growing regions. There can be no doubt that the European Green Deal with the aim of reducing the use of chemical pesticides by 50% by 2030 will have a significant stimulatory effect for both European and non-European researchers seeking to validate new combinations of BBR strategies described in this review. 

## Figures and Tables

**Figure 1 plants-11-03004-f001:**
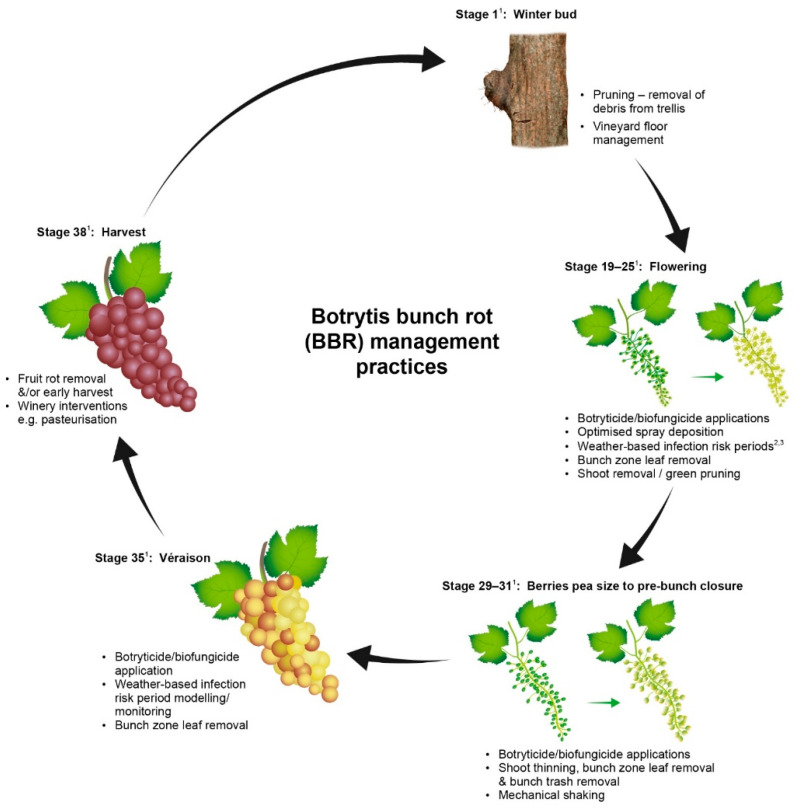
Phenological stage of the grapevine and botrytis bunch rot management options available to reduce risk of disease at harvest in New Zealand. (Superscript 1) This figure uses the modified Eichhorn-Lorenz phenological growth stages [38]. (Superscript 2,3) Examples of infection risk period modelling referred to in the figure are from Broome et al., 1995 [39] and Kim et al., 2007 [40].

**Table 1 plants-11-03004-t001:** Factors cited as the rationale for the urgent need for alternatives to synthetic pesticides and supporting literature.

Issue	Examples in the Literature
More restrictive governmental regulations such as European Union (EU) Directive 2009/128 and EU Green deal 2109 Farm to Fork Strategy	[8,9]
Increasing restrictions on allowable residues by export markets and global retailers	[10,11]
Older and less safe ingredients banned in the EU	[10,12]
Adverse environmental impacts including deleterious effects on non-target organisms (e.g., bees, beneficial insects, fish & birds)	[13,14,15]
Adverse effects on human health	[16,17,18,19]
Rapid emergence of resistance to AP ^1^, DC ^2^, HA ^3^, MBC ^4^, QoI ^5^, PPs ^6^ and SDHI ^7^ fungicides, including multiple resistance to several FRAC coded fungicide groups	[20,21,22,23,24]
Increasing demand for organics	[25]
Resurgence in interest in ‘Regenerative Agriculture’ in the last five years	[26,27]

^1^ AP (anilinopyrimidines), ^2^ DC (dicarboximides), ^3^ HA (hydroxyanilides), ^4^ MBC (methyl benzimidazole carbamates), ^5^ QoI (quinone outside inhibitors), ^6^ PPs (phenylpyrroles), and ^7^ SDHIs (succinate dehydrogenase inhibitors).

## Data Availability

Not applicable.

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
