# Peer review of "A Review of Cultural Practices for Botrytis Bunch Rot Management in New Zealand Vineyards"

_plants, 2022, doi:10.3390/plants11213004_

Round 1

Reviewer 1 Report

A Review of Cultural Practices for Botrytis Bunch Rot  Management in New Zealand Vineyards  

Authors: Dion Charles Mundy, Philip Elmer, Peter Wood  and Rob Agnew

In this review the authors have thoroughly put together scientific findings and research from different New Zealand wine growing regions with the aim to suppress successfully Botrytis Bunch Rot (BBR) by use of newly developed and traditional cultural control practices. With this review the authors show that there have been many efforts and research that BBR management in New Zealand vineyards can be successful by these new combinations of non-chemical BBR strategies with the purpose to reduce synthetic fungicides application in order to match the European Green Deal.

All in all the authors have done good work with this review. Some minor errors need to be clarified in the text (see below).

 Table 1: EU green deal 2019 (instead of 2109)

L 72: ‘a very significant achievement’ is too subjective for introduction and a review

L76: it would be informative to describe the NZ weather and vine growing conditions; are there some difference between the north and south? Also in Line 116 you mention ‘NZ wetter regions’, how is it in the other regions?

L78-83: in this section it is not clear whether you’re further describing the findings of Wurms et al. 2011 (29) or your own statement. Please make it more clear by writing down either the name of the authors (Wurms et al.)  or insert at the end of the sentence (L79-83) again the reference. Or if it is your statement please write ‘we’ instead of ‘the authors’ (as in Line 86) or make it more clear.

L97: please give here (Edwards et al. 2009) also a numbering as you do it in the rest of your text

L 125: ‘potential’ can be deleted

Figure 1: please add in the caption: in New Zealand

L 146: Please explain IP in the caption

L 169: Please write here some examples for the ‘different residues’ or write some sentences to this issue.

L177 – 178: is this last sentence in this paragraph your speculation or are any other sources you can refer to?

L 204: please insert here also the producer country

L 204: in this paragraph you refer to the authors, I assume you mean you as authors; is there any publication to the findings with air brush from 2007?

L 231: ‘site years’ with or without hyphen (see L 192); please write this word uniformly in the text

L 236: Please use the abbreviation LR in the whole text and explain it at the beginning, for example in line 215

L 307-309: ‘Table grapes have used .... post-harvest.’ This sentence sounds weird

L319: Could you please write here instead of ‘published in 2021’ the name + reference number like Mundy et al. [77]

L 336: Please mention the reference of the ‘2021 publication’, since it is unclear whom you’re referring to

L 337: which is the reference of the berry assay? 77 or 81; please write it down since it is confusing.

L 340: why a new pargraph? Doesn’t it refer to the same berry assay?

L 340-344: What are those temperatures and humidity? Data about the ideal conditions, like temperature, humidity and spore amount would be informative here

L 347: ‘reported in 2021’ can be deleted.

L349-351: ‘…low incidence means that there were some degree of disease occurrence although the berries were not inoculated’. Where was then the source of infection for the low incidence when you assume that latent infection was not present?

L 355: ‘…primary metabolism of green berries….’

L 372-374: In the end of this paragraph your statement gets a bit confusing. Can you please write the reference for the findings of 50 % disease reduction; I’m wondering whether 50% disease reduction can be compared with the berry susceptibility assay achieving 20 % reduction in the susceptibility at artificial conditions.

L395: It is easier for the reader when you mention the authors instead of telling Spanish study; just write as example: Tardaguila et al. [74] were interested in … (the same applies to L 398; L399).

L396-397: actually, yield is not physiological measurement; maybe you can write ‘….effects of thinning on physiological measurements and yield.’

L 400: Please insert ‘disease’ before incidence.

L435: ‘…, wine grape varieties’ can be deleted

L 483: which authors do you mean here?

L 505: within

L 527: B. cinerea - in italics (also in Lines 560; 618)

L 546-552: is there a reason for writing in italics?

L555-557: Can you name the problems?

Author Response

Thank you for your comments and suggestions. we have made most of the changes but have kept Leaf removal as words rather than LR as also suggested by one of the other reviewer.

Reviewer 2 Report

It is a comprehensive review of updated knowledge on this subject.

Just a few minor comments: 

Line 143- Figure 1: write "leaf removal" instead of "LR" in the graphic.

Lines 546-552: correct the italic font.

Lines 833-834: A 2012 paper, still in press? Please, correct publication data.

Author Response

thank you for your comments an we have made the suggested changes.

Reviewer 3 Report

The review manuscript  "A review of cultural practices for Botrytis bunch rot management in New Zealand vineyards" is well written and it contains both previously knowledge related to BBR management practices, as well as some new knowledge on this subject and has high value both for the scientific community and for producers. It also defines lacking pieces of knowledge that should be investigated in the future. 

Author Response

Thank you for your comments.
